:ᐧPLOS | ONE

# Movement and habitat selection of the western spadefoot (*Spea hammondii*) in southern California

**Katherine L. Baumberger**[1¤a]*, **M. V. Eitzel**[2¤b], **Matthew E. Kirby**[3], **Michael H. Horn**[4]

**1** Environmental Studies Program, California State University at Fullerton, Fullerton, California, United States of America, **2** Department of Environmental Science, Policy, and Management, University of California at Berkeley, Berkeley, California, United States of America, **3** Department of Geological Sciences, California State University at Fullerton, Fullerton, California, United States of America, **4** Department of Biological Science, California State University at Fullerton, Fullerton, California, United States of America

¤a Current address: Western Ecological Research Center, United States Geological Survey, Santa Ana, California, United States of America
¤b Current address: Science and Justice Research Center, University of California at Santa Cruz, Santa Cruz, California, United States of America
* kbaumberger@usgs.gov

**Data Availability Statement:** All data files are available from the Dryad database (doi:10.5061/dryad.8359820, https://datadryad.org/review?doi=doi:10.5061/dryad.8359820).

**Funding:** K. B. received funding from the Laguna Canyon Foundation (https://lagunacanyon.org/), the Association of Zoos and Aquariums Amphibian Taxon Advisory Group Conservation Fund (https://www.aza.org/amphibian-conservation), and the Natural Communities Coalition (formally the Nature Reserve of Orange County) (https://occonservation.org/). The funders had no role in

## Abstract

Agricultural activity, urban development and habitat alteration have caused the disappearance of the western spadefoot (*Spea hammondii*) from 80% of its geographic range in southern California. Despite the western spadefoot's continuing decline, little research has been conducted on its natural history. The home range of adult spadefoots is unknown, and their use of upland habitat is poorly understood. Both factors are important for the long-term conservation of the species because adult spadefoots spend the majority of their lives away from breeding pools in self-excavated burrows. Between January 2012 and January 2013, we surgically implanted radio transmitters in 15 spadefoots at two locations and recorded their movements and habitat use. The mean distance moved between burrow locations was 18 m (SD ± 24.1 m, range 1–204 m). The mean distance of burrows from the breeding pools was 40 m (SD ± 37.42 m, range 1–262 m). Rain was a significant predictor of spadefoot movement, with more rain predicting higher probability of movement and larger distances moved. At remote sensing scale (1 m) spadefoots selected grassland habitat for their burrow locations. At the microsite scale (< 1 m) spadefoots strongly selected duff over grass or shrub cover. Spadefoots burrowed in friable, sandy/loam soil with significantly less clay than random pseudoabsence points. This research enhances our understanding of a little-studied species and will contribute to the development of effective management plans for the western spadefoot.

## Introduction

Habitat loss is one of the main causes of amphibian decline throughout the world [1–4]. Because of this loss, amphibian conservation has historically been focused on the preservation or restoration of aquatic habitats [5], and on the upland habitat use of adult amphibians [6–7].

study design, data collection and analysis, decision to publish, or preparation of the manuscript.

**Competing interests:** The authors have declared that no competing interests exist.

Specifically, attention has been given to determining the minimum buffer around a breeding pool required to conserve the adult population [8–11]. For a wide variety of amphibian species, the mean core terrestrial habitat ranges between 205–368 m from the edge of aquatic habitat [12]. Conservation plans benefit strongly from specific information about particular species of concern, and determining adequate terrestrial buffers for these species is key to ensuring the viability of their populations [5]. Delineating the upland habitat use of fossorial anurans is essential given that they spend the majority of their lives away from breeding pools in terrestrial burrows. Defining a required buffer around the pool is particularly important in conservation planning for these species; distances from breeding pools at which fossorial anurans aestivate can be as large as 370 m to 2,350 m, e.g. for the Great Basin spadefoot (*S. intermontane*) and eastern spadefoot (*S. holbrookii*) [13–15]. In addition to the size of the buffer, restoration planning requires information regarding types of land cover preferred by the species of concern.

We examined use of upland habitat by the western spadefoot (*Spea hammondii*), a burrowing anuran that has been extirpated from 80% of its range in southern California because of agricultural expansion and urban development [16–17]. This extensive habitat loss has led to the western spadefoot's listing as a California Species of Special Concern [18], and the species is under review for listing as Endangered or Threatened under the Endangered Species Act [19]. The western spadefoot is endemic to California and historically inhabited lowlands such as river floodplains and washes in the Central Valley and along the coast from central California to northwestern Baja California [17,20]. Adult spadefoots spend most of their time underground and emerge primarily on rainy nights to feed and to breed in vernal pools [21–22]. The home range size of adult western spadefoots is unknown, and the maximum distance moved from breeding pools has not been established [18,23].

The objective of this baseline study was to confirm the movement ecology and determine the habitat use of spadefoots at two sites in Orange County, California. We used radio-telemetry to monitor 15 spadefoots to establish a baseline understanding of four aspects of their habitat use patterns: 1) basic movement ecology, including home range size and distances moved away from breeding pools; 2) identification of variables predicting movement; 3) vegetation characteristics at burrow locations (from remote sensing classification and from vegetation surveys of the 1 x 1 m area encompassing the spadefoot's position); and 4) soil characteristics of aestivation locations (defined as a residence exceeding three weeks in the same burrow location). We were thus able to contribute to basic understanding of the movement and habitat selection of the western spadefoot which can support future studies and management planning.

## Materials and methods

### Ethics statement

This study was conducted under California Department of Fish and Wildlife permit SC-07437 and approved by the Institutional Animal Care and Use Committee at California State University, Fullerton (Protocol No. 11-R-05). Animal handling followed the Herpetological Animal Care and Use Committee of the American Society of Ichthyologists and Herpetologists guidelines [24]. Animals were anaesthetized via immersion in MS-222 (0.4g dissolved in 500ml of water) at a veterinary clinic, and every effort was made to minimize suffering.

### Study area

This study was conducted in two protected parks, Crystal Cove State Park (UTM 11S 423775 E 3714365) and the Laguna Coast Wilderness Park (UTM 11S 429081 E3714390) in Orange

County, California. Together, these two areas form the largest remaining contiguous parcels of coastal sage scrub habitat in Orange County. These sites are also home to some of the few remaining vernal pools in the area. The site at Crystal Cove State Park is in a campground with greater potential for human nighttime activity. The site in the Laguna Coast Wilderness is a short drainage ditch along a popular hiking trail. Both areas were closed for several days following rain events to prevent visitors from damaging the wet trails. These closures reduced the impact of human interference on spadefoot movement by limiting the usual heavy human traffic at the study sites. All necessary permits were obtained from both Crystal Cove State Park and Orange County Parks, and our study complied with all relevant regulations.

The area of the Crystal Cove State Park breeding pool was approximately 38 m$^2$, while the Laguna Coast Wilderness pool was 3 m$^2$. However, during the 2011–2012 breeding season the Crystal Cove pool did not hold water. The Laguna Coast pool only held water for a week. The total precipitation during our study period (20.4 cm) was well below the 60-year average for the area (mean = 32.7 cm, SD ± 17.1). The average temperature for January through December 2012 (mean = 16.7˚ C, SD ± 5.5) was similar to the 60-year average (mean = 16.2˚ C, SD ± 1.29) [25].

## Radio telemetry

During the potential breeding period, between the end of January and the end of April 2012 [16], we opportunistically captured a total of 15 spadefoots (see S1 Table for capture dates). Seven of the 15 animals were caught in Crystal Cove State Park: these spadefoots were most likely on the surface foraging because the breeding pool did not fill. The eight animals caught in the Laguna Coast Wilderness were found in or near the breeding pond while water was present. Only three female spadefoots were captured, all at Crystal Cove State Park. The animals were sexed in the field based on the presence/absence of nuptial pads [17]. The sexing was confirmed during surgery, as all three females had eggs present. We surgically implanted small radio transmitters (Model A2455, Advanced Telemetry Systems, Isanti, MN) into the coelomic cavities of each animal using the methods of Timm et al. [13]. Surgical implantation was deemed to be safer and more reliable for a fossorial species because external attachment of transmitters can cause abrasions and entanglement [24,26]. The transmitters weighed between 1.1 g and 1.2 g, putting them at less than 5% of the spadefoot's weight (mean 32 g ± 5, range 26–40 g) (Table 1) [27]. We excavated the spadefoots from their burrows one time during the study to perform a welfare check approximately a week after their surgeries, and no ill effects (e.g. redness/swelling at incision site, weight loss) were observed.

**Table 1. Summary of spadefoot characteristics and movement.**

| Number of spadefoot | 15 Total | 3 Female | 12 Male |
|---|---|---|---|
| | **Mean** | **SD** | **Range** |
| **Snout-vent length (mm)** | 62 | 5 | 54–70 |
| **Mass (g)** | 32 | 5 | 26–40 |
| **Capture date** | 2012/03/20 | 39 days | 2012/01/21–2012/04/26 |
| **Last located date** | 2012/12/18 | 32 days | 2012/10/13–2013/01/21 |
| **Number of locations** | 36 | 14 | 22–59 |
| **Number of burrows used** | 13 | 9 | 4–35 |
| **Maximum distance from pool (m)** | 69 | 60 | 16–262 |
| **Home range size (MCP m$^2$)** | 1340 | 1690 | 25–5620 |
| **Mean distance between burrows (m)** | 18 | 12 | 9–57 |
| **Depth of burrow from welfare check (m)** | 0.1 | 0.05 | 0.01–0.18 |

We tracked the animals with a three-element Yagi antenna and a portable receiver (model TR-4, Telonics, Mesa, AZ) two times a week from January to June 2012. From July through October 2012, we monitored the animals' aestivation locations every other week. From October 2012 through January 2013, we monitored the animals once a week until the batteries gave out on the transmitters. We recorded burrow locations with a hand-held GPS receiver (model Rino 520, Garmin International, Inc., Olathe, KS) and then uploaded the points to a Geographic Information System (GIS) (ArcGIS 10 and 10.1, ESRI, Redlands, CA).

## Vegetation characteristics

To document vegetation characteristics at burrow locations, we placed a 1 m x 1 m Polyvinyl Chloride (PVC) square centered at the burrow opening. Within the square, we visually determined percent cover of five vegetation classes: grass, forbs, shrubs, leaf litter (recently fallen leaves), and duff (dead and decomposing vegetation from previous seasons) [28] to the nearest 5 percent. We also recorded topographic slope and aspect to the nearest degree using a hand-held compass with a built-in clinometer (Brunton, Riverton, WY). See Table 2 for a summary of these variables. In October 2012, we took the same measurements at 102 random pseudoabsence locations generated with ArcGIS and Geospatial Modeling Environment 0.7.2.1 (GME) [29]. The points were within a 300 m radius of the breeding pool (so as to encompass the maximum single movement of our tracked spadefoots), and at least 1 m away from a known spadefoot burrow [30].Though there is some temporal mismatch between these pseudoabsence points and the known spadefoot points, we carefully distinguished between dead grasses and forbs from 2012 and duff from 2011 or earlier. The points were stratified proportionally between the two habitat types present at our sites, "grassland" and "shrub," as classified from heads-up digitization of a sub-meter spatial resolution satellite image (ESRI base map, sources: ESRI, DigitalGlobe, GeoEye, i-cubed, USDA, USGS, AEX, Getmapping, Aerogrid, IGN, IGP, swisstopo, and the GIS User Community). We chose to stratify proportionally between these two habitat types because we did not have previous knowledge regarding which type the spadefoots would select and wanted to ensure adequate representation of both habitat types.

**Table 2. Summary of characteristics at burrow locations and rainfall data for the winter of 2011–2012.**

|  | Mean | SD | Range |
|---|---|---|---|
| Percent sand | 52 | 15 | 29–82 |
| Percent silt | 38 | 11 | 13–53 |
| Percent clay | 10 | 4.6 | 3.8–18 |
| Total organic matter (percent) | 4.6 | 1.2 | 2.4–7.1 |
| Topographic slope (percent) | 6 | 6.5 | 0–30 |
| Aspect: northness | -0.22 | 0.72 | -1.0–1.0 |
| Aspect: eastness | 0.02 | 0.66 | -1.0–1.0 |
| Percent grass cover | 25 | 28 | 0–100 |
| Percent shrub cover | 24 | 35 | 0–100 |
| Percent leaf litter cover | 15 | 33 | 0–100 |
| Percent forb cover | 15 | 20 | 0–100 |
| Percent duff cover | 40 | 36 | 0–100 |
| Duff height (cm) | 2.5 | 3 | 0–28 |
| Vegetation height (cm) | 45 | 36 | 0–150 |
| Locations with other animal burrows | 72/167 | NA | NA |
| Number of days with rainfall | 29 | NA | NA |
| Amount of rainfall (cm) | 0.41 | 0.89 | 0–4.5 |

## Soil characteristics

We followed the methods of Kirby et al. [31] to measure the percent total organic matter, percent silt, percent clay, and percent sand of aestivation locations and pseudoabscence points. We used a core sampler to collect soil from 11 spadefoot burrows (this represents all 15 spadefoot aestivation burrows because several spadefoots aestivated within 1 m of each other) and 10 random points stratified spatially from our list of random points described above. The cores averaged 21.43 cm in length (range = 15–29 cm). This depth was consistent with the depth of burrows observed during welfare checks of the tracked animals (see Table 1). Soil cores were subsampled every 2 cm for compositional analysis. These analyses included determining grain size and using the loss-on-ignition method to obtain percent total organic matter [32]. Grain-size was measured on a Malvern Mastersizer 2000 laser-diffraction grain-size analyzer coupled to a Hydro 2000G. All data were reported as percent by volume. Grain-size data were classified using Wentworth's [33] classification, dividing clay at < 3.9 μm, silt at 3.9–62.5 μm, and sand at 62.5–2000 μm [34]. We averaged soil characteristics for the entire length of each core because spadefoots presumably moved through the entire profile to reach final placement in their burrows.

## Data analysis

We used ArcGIS and GME to estimate home range as a 95% minimum convex polygon (MCP) for each spadefoot and for each site. We estimated the utilization distribution (UD) via the a-local convex hull (a-LoCoH) method [35–36] using the rhr package [37] in R [38]. The outputs from the UD analysis capture use of space at each site, rather than individual home ranges [36]. We pooled data by site because of the low number of locations observed.

We used R for all statistical models that follow. Unless otherwise noted, we calculated p-values using likelihood ratio (LR) tests, comparing a full model including all variables with a reduced model eliminating the variable of interest. LR tests are more conservative than t-tests and are less sensitive to unbalanced designs [39]. Where models included only fixed effects, we used "glm" to fit models for LR tests and to obtain parameter estimates. Where models included a random effect (movement models), we used glmmADMB [40] to fit the generalized linear mixed models.

To evaluate the spadefoots' selection of burrowing location characteristics, we analyzed two scales of habitat classification: site scale, based on visual classification of remotely-sensed imagery; and microsite scale, based on the vegetation quadrats. At the site scale, we conducted a Chi-squared test of independence between spadefoot presence/pseudoabsence and vegetation type (grassland versus shrub). At the microsite scale, we fit four separate binomial generalized linear models for spadefoot presence or pseudoabsence as predicted by: 1) each of the five vegetation classes, 2) site, vegetation height, duff depth, 3) slope and aspect (transformed into 'northness' = cos (aspect) and 'eastness' = sin (aspect)), and 4) the effect of other animal burrows (e.g. gopher and ground squirrel) on spadefoot presence. The models for these sets of burrow location characteristics were separate because the sample sizes did not match (sample sizes for each set of models are listed in Results tables). Note that we did test the study site (Laguna Coast Wilderness versus Crystal Cove State Park) in these models, but as it was never significant, it was not included in the results tables. This outcome is unsurprising as we designed the study for equal numbers of spadefoots at each site.

To evaluate the spadefoot's soil preferences at aestivation locations, we used binomial generalized linear models for spadefoot presence or pseudoabsence predicted by soil characteristics: soil texture (percent sand, silt, and clay) and total organic matter. At our sites, soil composition was confined to a narrow combination of the three textures, and in addition to

the fact that percent sand, percent silt, and percent clay sum to 100%, these percentages correlated strongly (sand/clay: -0.90, sand/silt: -0.98, silt/clay: 0.80). Therefore we conducted a principal components analysis (using the "prcomp" function in R), which gave a first component reflecting the greatest variation in soil composition in our sites (98%), and a second component orthogonal to the first which reflected the second greatest variation in soil characteristics (2%). The third component reflected the degree to which the three textures did not sum to 100 percent, i.e. the analysis error. As this component accounted for very little variation (0.004%), we did not include it in the binomial model for spadefoot presence/pseudoabsence. Because principal components can be difficult to interpret, we also report the parameters for a binomial model with presence/pseudoabsence predicted by silt and clay (the least correlated of the three textures). We created a similar generalized linear model for spadefoot presence or pseudoabsence predicted by total organic matter. Organic matter and soil texture variables were tested separately because of differing sample sizes.

To determine drivers of spadefoot movement, we used a "hurdle" model [41] in which we first modeled the probability of movement (binomial generalized linear model) and then separately modeled the distances moved (gamma generalized linear model with a log link). Predictors in both models included rainfall in centimeters [42], sex of the spadefoot, phase of the moon [43], and study site (all as fixed effects), and individual spadefoot as a random effect. These models were fit using glmmADMB. Note that the sampling effort at the two sites was not equal, as the spadefoots were tagged much earlier at Laguna Coast Wilderness and therefore the number of possible movements was higher. This difference in sampling effort was less important for the habitat models, but for the movement models it was critical that we control for site.

## Results

The transmitters lasted an average of 272 days (range 224–335 days). During that time, we fixed the location of the spadefoot burrows 532 times, representing 195 unique spadefoot burrows. The maximum distance the spadefoots were found from the pools ranged from 16 to 262 m (Table 1, S1 Table), with a mean maximum distance of 69 m ± 61.48. The spadefoots used a mean of 13 burrows (SD ± 8.5), and the mean distance between burrow locations was 18 m (SD ± 24.2). They used 4–31 unique burrow sites (mean 11 ± 7.8) during the study. Nine of the 15 spadefoots (60%) reused one or more burrows at least once after moving to a different burrow. Outside of their aestivation period, the spadefoots shifted their burrow location an average of every 8 ± 7 days, and 147 of 194 (~76%) movements between burrows were ≤ 25 m. Spadefoots began aestivating in May and June 2012. The spadefoots remained in their aestivation burrows 125–220 days (mean 157 days ± 24.5), with the last spadefoot moving to a new burrow from its aestivation site on December 21, 2012. The 95% MCP home range size for Laguna Coast Wilderness was 8,242 m$^2$, larger than for Crystal Cove State Park, which had a 95% MCP size of 6,285 m$^2$. The UD for the Laguna Coast Wilderness was also larger with an area of 1115 m$^2$. The UD for Crystal Cove was 599 m$^2$, roughly half (53%) of the UD of Laguna Coast Wilderness (Fig 1). All the spadefoots were presumed to be alive at the end of the study, based on movement before transmitter failure. We observed no predation events or injuries. We did not remove the transmitters because of the potential negative consequences of removing transmitters that had been encapsulated by connective tissue [44].

### Drivers of movement

Although spadefoots did move when no rain was present (Fig 2), rain significantly predicted spadefoot movement, as did the animal random effect (Table 3). For an average animal, the

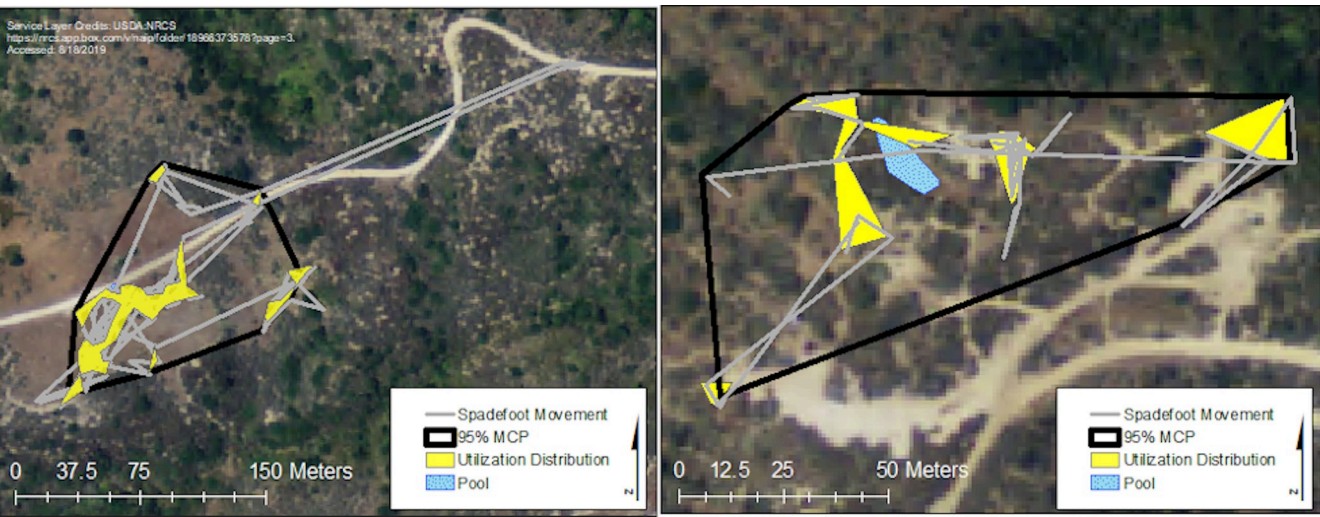

**Fig 1. Home range and utilization distribution of spadefoots.** Home ranges (black outline) represented as 95% minimum convex polygons, and utilization distribution (yellow polygons) of adult western spadefoots in Orange County, California, USA, in 2012 for Laguna Coast Wilderness (left) and Crystal Cove State Park (right). Blue solid polygons show the breeding pool at each site.

model predicts a 46 percent chance the spadefoot will move with no rainfall (not significantly different than 50 percent), while at maximum rainfall (4.5 cm) the model predicts an 87 percent chance that the animal will move. We also found significantly more movement at Laguna Coast Wilderness, consistent with that site's longer sampling period. Number of meters moved away from or towards the breeding pool was predicted by rain in cm and by the animal random effect. With no rainfall, the model predicts that the average animal will move 21 m, while at maximum rainfall the model predicts that the average animal will move 163 m.

## Characteristics of burrow locations

Duff depth and vegetation height were not significant predictors of spadefoot presence (Table 3). At the site scale (from the imagery-classified vegetation types), spadefoots strongly selected for grassland rather than shrubs (Fig 3F, Table 3). At the microsite scale, spadefoot strongly selected duff over grass or shrub cover (Fig 3A–3E): the model predicts that a spadefoot encountering a site composed entirely of grass has probability 0.30 of choosing to burrow there (significantly less than 0.50); if the site were composed entirely of shrub, a probability 0.36 of choosing to burrow there (significantly less than 0.50); and if it were composed entirely of duff, a probability 0.94 of choosing to burrow (significantly more than 0.50). No other cover types were significant, including open ground (Table 3).

The model predicts that spadefoots encountering sites without burrows created by other animals (e.g. gophers and ground squirrels) have a probability 0.43 of choosing to burrow there (not significantly different from 0.50), whereas a spadefoot encountering a site with a pre-existing burrow has probability 0.76 of choosing to burrow there (Table 3). Spadefoots selected burrows on flatter slopes with south-eastern aspects. For example, the model predicts that, for a spadefoot encountering a site on a south-facing slope, there would be a 0.02 probability that they would choose to burrow if the site had a 30 degree slope, a probability 0.62 if it had a 6 degree slope (the mean slope at our sites), and a 0.90 if it was on flat ground. If the spadefoot encountered a site on a 6-degree slope, they would have a 0.49 probability of choosing to burrow if the site was a north-facing location, a 0.43 probability if it were west-facing, and a 0.78 probability if it were east-facing (Table 3).

## Individual Spadefoot Movement and Rainfall Events

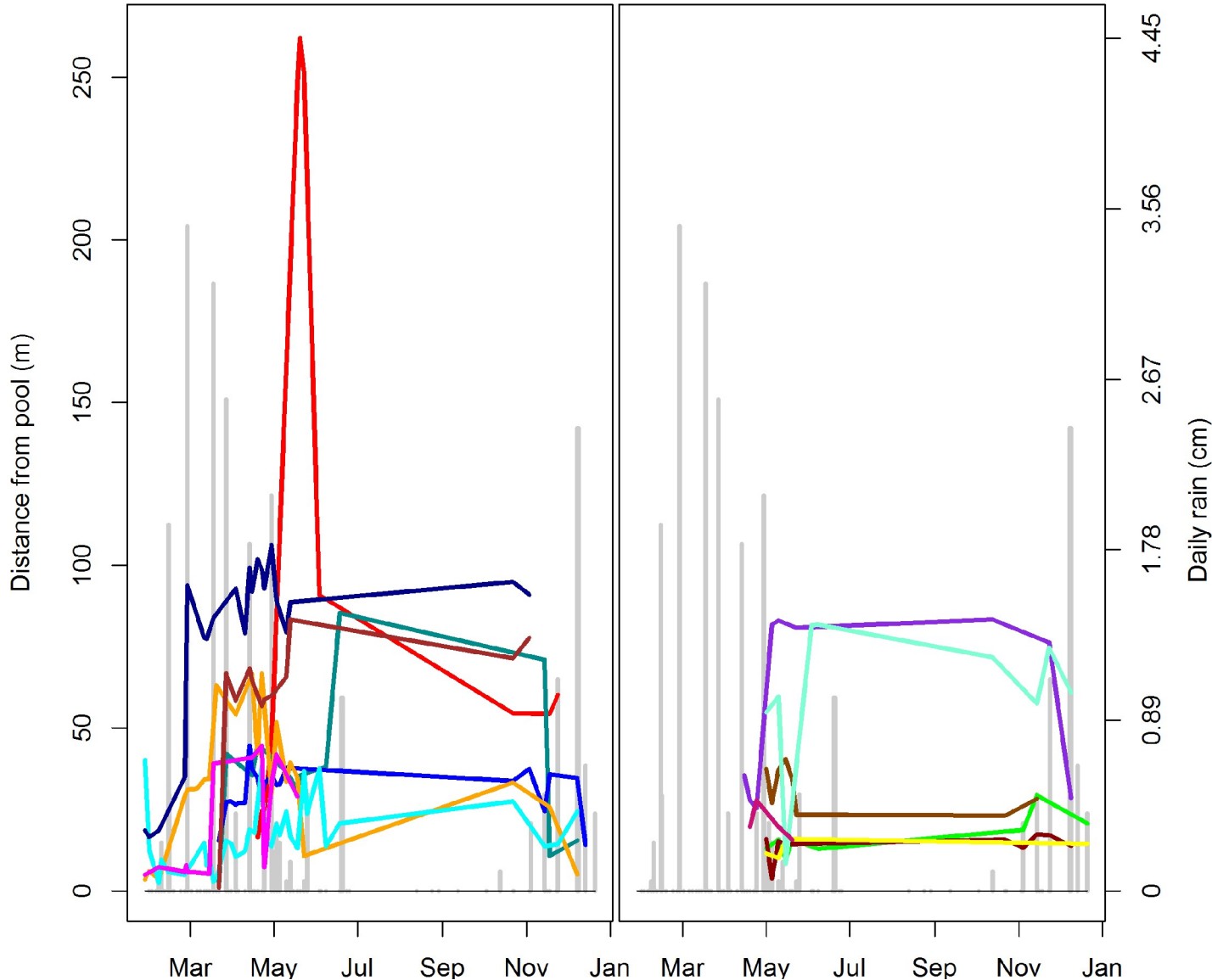

**Fig 2. Spadefoot movement away from breeding pools vs. rainfall.** Spadefoot movement away from breeding pools over the course of the study (January 2012 through January 2013) at A) Laguna Coast Wilderness and B) Crystal Cove State Park. Daily rainfall in Orange County, USA, is displayed in gray bars in the background, and each line represents an individual spadefoot's distance from the breeding pool throughout the year. Most individuals remained within 100 m radial distance from the breeding pool, while one individual went much farther. Movements occurred outside rainfall events, and movement did not always occur during rainfall events, but statistically the relationship is significant (see Table 3). Individuals were tracked for a shorter time period at Crystal Cove State Park.

### Soil characteristics of aestivation locations

Total organic matter was not a statistically significant predictor of whether a spadefoot would choose to aestivate at a given site. The soil-texture classification for all soil cores ranged from loamy sand to silt loam (Fig 4). The first principal component, representing the greatest variation

**Table 3. Model untransformed parameter estimates, standard errors, and statistical significance.**

| Model | N | Parameter | Estimate | SE[1] | P-value |
|---|---|---|---|---|---|
| **Burrows: site level[2]** | 297 | Shrub vs. Grassland | 36.2, df = 1 | NA | <0.001 |
| **Burrows: vegetation structure** | 271 | Vegetation height | NS | NS | NS |
| | | Duff height | NS | NS | NS |
| **Burrows: vegetation cover** | 300 | Percent duff | 0.02 | 0.006 | 0.002 |
| | | Percent shrubs | -0.01 | 0.006 | 0.02 |
| | | Percent forbs | NS | NS | NS |
| | | Percent grass | -0.02 | 0.007 | 0.02 |
| | | Percent leaf litter | NS | NS | NS |
| | | Percent open ground | NS | NS | NS |
| | | Percent tree | NS | NS | NS |
| **Burrows: physiography** | 285 | Slope | -0.2 | 0.03 | <0.001 |
| | | Aspect: northness | -0.55 | 0.22 | <0.001 |
| | | Aspect: eastness | 0.79 | 0.24 | 0.01 |
| **Burrows: other burrow** | 167 | In other animal burrows | 1.42 | 0.35 | <0.001 |
| **Aestivation: soil texture** | 20 | Principal component 1 | NS | NS | NS |
| | | Principal component 2 | -2.79 | 1.55 | 0.03 |
| | | Percent clay[3] | -0.46 | 0.25 | 0.03 |
| | | Percent silt[3] | 0.17 | 0.1 | 0.05 |
| **Aestivation: organic matter** | 19 | Total organic matter | NS | NS | NS |
| **Movement: yes/no** | 532 | Rainfall (cm) | 0.46 | 0.11 | < 0.001 |
| | | Mean site difference | -0.81 | 0.35 | 0.02 |
| | | Phase of moon | NS | NS | NS |
| | | Sex of spadefoot | NS | NS | NS |
| | | Animal random effect | 0.33 | NA | 0.03 |
| **Movement: distance (m)** | 181 | Rainfall (cm) | 0.13 | 0.06 | 0.02 |
| | | Mean site difference | NS | NS | NS |
| | | Phase of moon | NS | NS | NS |
| | | Sex of spadefoot | NS | NS | NS |
| | | Animal random effect | 0.41 | NA | < 0.001 |

[1]Standard errors for random effect variance components are not symmetrical and not included.

[2]Chi-squared test of independence.

[3]Separate model from principal components analysis.

in soil textures across our sites, was not significant in determining spadefoots' choice of aestivation sites, but the second component was significant (Table 3). Because the second principal component had weightings of -0.21 for sand, -0.58 for silt, and 0.79 for clay, and the corresponding model parameter was negative (-2.79), we conclude that spadefoots tended to prefer soils with less clay and more sand and silt, with the preference against clay as the more dominant effect (Fig 4). This agrees with the results of a model including just percent clay and percent silt: for a site with one additional percent clay from the mean value, the clay/silt model predicts that the probability of a spadefoot choosing to aestivate there would drop from 0.51 to 0.40, while for an additional percent silt, the probability of spadefoot aestivation would rise from 0.51 to 0.56.

## Discussion

We documented details of the terrestrial activity and burrow site characteristics of *S. hammondii*, which was previously unknown for the species [18,23]. No previous studies had been

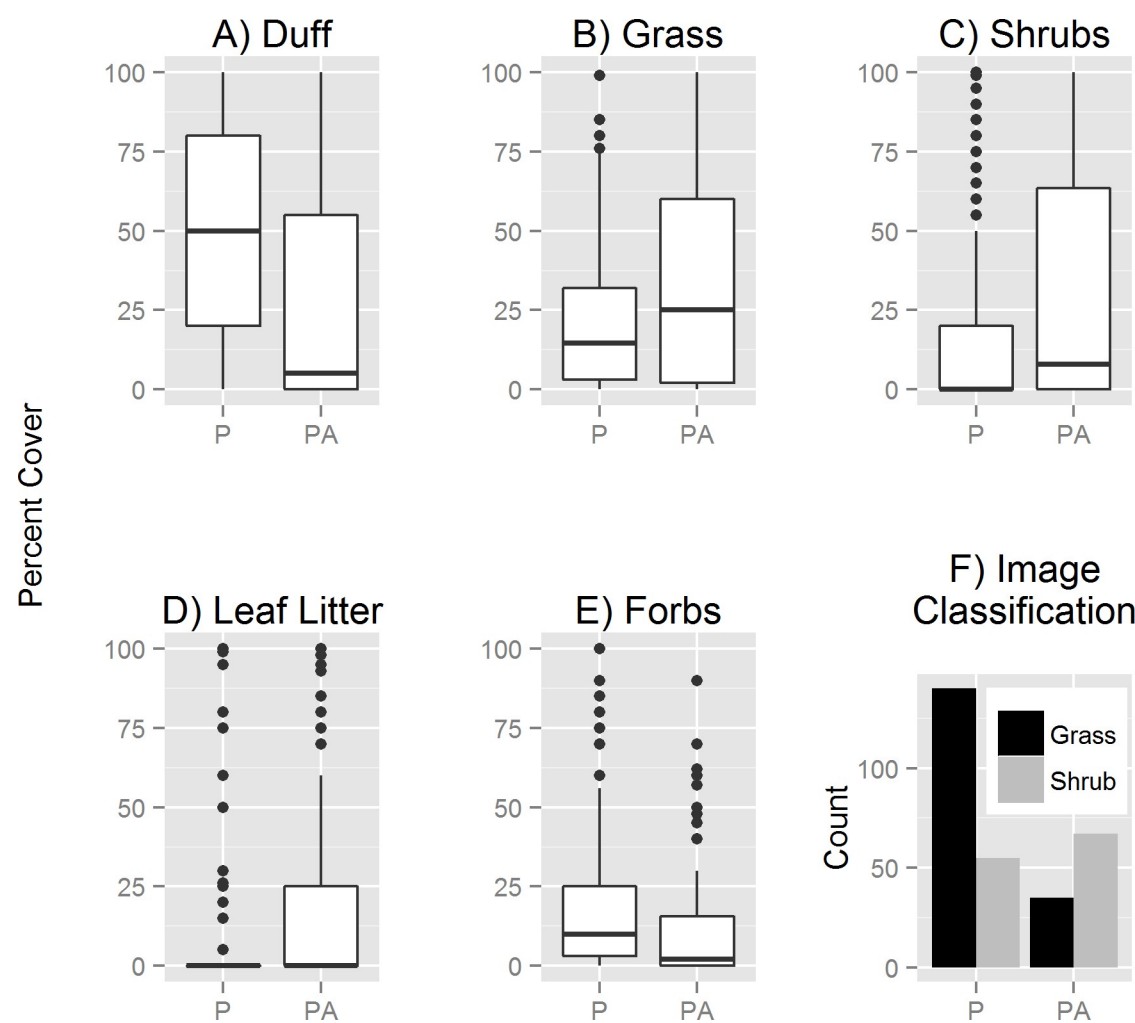

**Fig 3. Habitat characteristics of spadefoot burrow locations.** Habitat characteristics within 1 m$^2$ of spadefoot burrow locations (P) and pseudoabsence (PA) locations in Orange County, California, USA, in 2012. Percent cover of A) duff, B) grass, C) shrubs, D) leaf litter, and E) forbs, in addition to F) the classification from high spatial resolution imagery of each presence or pseudoabsence point as "grass" or "shrub." (Two other cover classes are not shown: tree cover, which is rare at our sites, and open ground, which is the complement to grass, leaf litter, and duff: they sum to 100%).

conducted on the upland habitat use of western spadefoots; the work done by Ruibal et al. [45] focused on *Spea multiplicata* when it was still thought to be a subspecies of *S. hammondii*. Our findings can establish minimum buffer distances and type of habitat required for the conservation of the species and set a starting point for future study of this species' habitat needs–key information given that habitat loss is the driving factor in the western spadefoot being considered for listing under the Endangered Species Act [19] as well as its status as a Priority I Species of Special Concern in California [18].

Current conservation efforts for the western spadefoot include delineating buffer zones around known breeding pools. Based on our results, the 76 m buffer around vernal pools required by the California Department of Fish and Wildlife [46] encompasses 169/194 (87%) of spadefoot burrows. This amount may be enough habitat to maintain the local populations at our sites [12], but it does not consider habitat quality within the buffer area. The minimum terrestrial buffer distance of 368 m recommended by the U. S. Fish and Wildlife Service [15] would encompass all the spadefoot home ranges in our study. In rapidly urbanizing

# Spadefoot Soil Texture

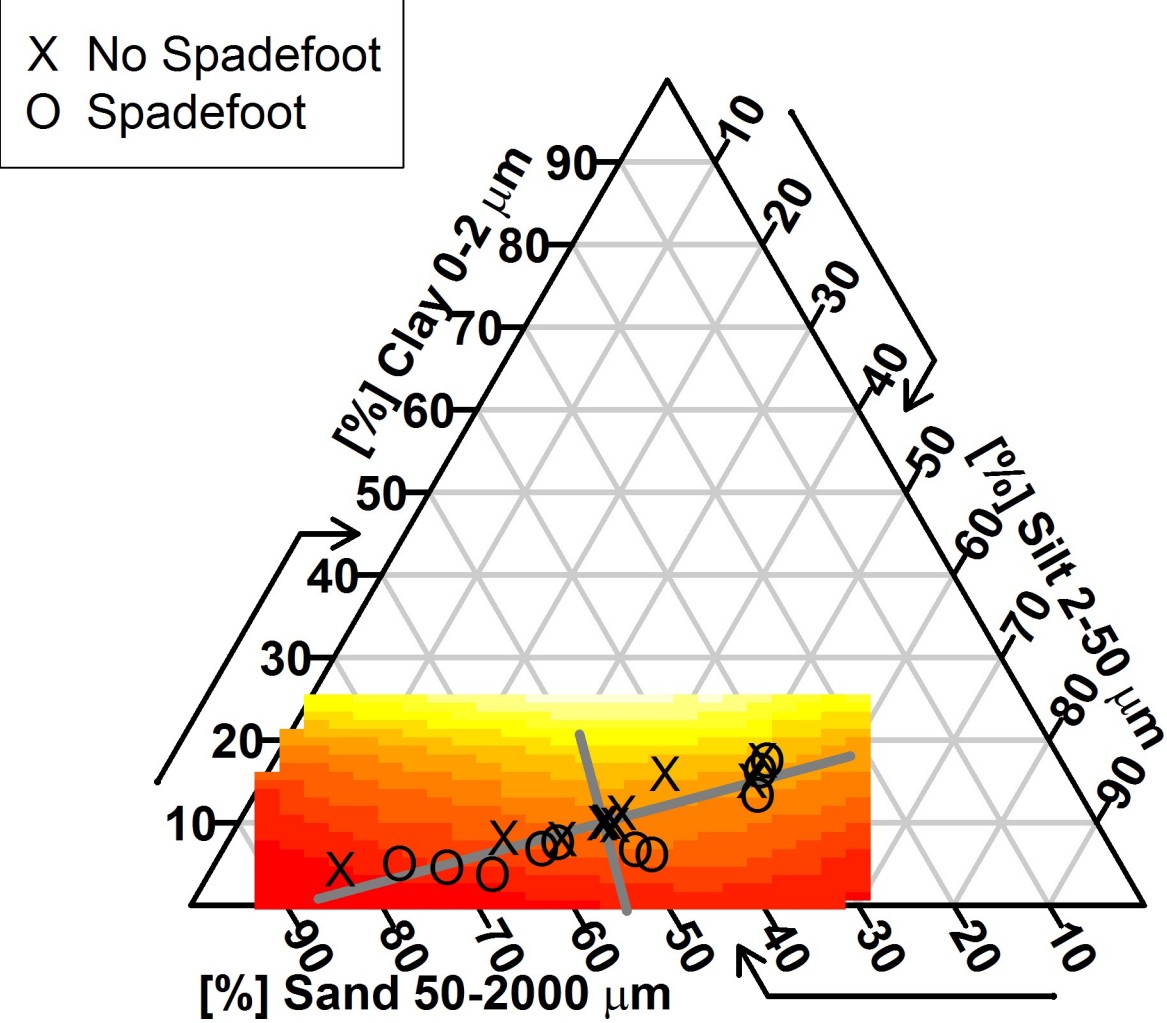

**Fig 4. Ternary soil map showing percent composition of clay, sand, and silt at spadefoot sites.** On each side the arrow indicates which direction to read for that soil type (e.g. "clay" percentages are read horizontally). Black "X"s indicate the soil texture of pseudoabsence sites and black "O"s indicate soil texture of aestivation sites for our 15 spadefoots. The thick gray lines indicate principal component combinations of soil textures: principal component 1 (longer line) reflects the greatest variation in soil composition in our sites (which spadefoots did not respond to in choosing aestivation sites), and principal component 2 (shorter line) reflects the deviation from the first component (which spadefoots did respond to in choosing aestivation sites). In particular, spadefoots tend to avoid clay soils. In color are model predictions using those two principal components: red signifies 99.7% probability of finding a spadefoot, while light yellow indicates 79.6% probability of finding a spadefoot.

environments, implementation of a 368 m buffer around all breeding pools might not be feasible; therefore, important habitat elements such as aestivation locations and migration/dispersal corridors should be identified and conserved [30].

For the *S. hammondii* in coastal California, one important habitat element is grassland for burrowing, as indicated by our satellite-imagery based analysis. Other spadefoot species including *Spea intermontana* and the European spadefoot (*Pelobates fuscus)*, were found to use

grassland or areas with short vegetation rather than shrub habitat; however, both of those species were more likely to burrow in bare ground [14,26], unlike the *S. hammondii* at our sites which had over half of their burrows in areas characterized by duff. The duff present at both sites is the product of non-native annual grasses in the genera *Bromus* and *Avena*; very little native grass was present at either site. The duff could act as cover for spadefoot movement and could also limit evaporation, thus conserving soil moisture for spadefoot burrows [47]. In addition, three spadefoots aestivated within one meter of each other at the only tree-dominated habitat at one site. The tree's shade could keep the area cool during a hot summer, and it may also regulate soil moisture through leaf-litter mulch and hydraulic lift of the roots [48]. Tree cover could therefore be another important habitat element to conserve, and further study is warranted.

Another major habitat element may be the presence of mammal burrows. Like the sympatric California tiger salamander (*Ambystoma californiense*), the western spadefoot showed a preference for burrow placement adjacent to or in California ground squirrel (*Spermophilus beechyi*) and Botta's pocket gopher *(Thomomys bottae)* burrows [6]. The benefit of using mammal burrows include the ease of digging and the potential for optimal moisture conditions [49–50]. However, unlike salamanders, spadefoots are excellent diggers; therefore, we cannot assume that they were inside the gopher and ground squirrel burrows. In fact, during welfare checks, we found that one spadefoot had dug his own burrow adjacent to, but not in, a ground squirrel burrow. Use of pre-existing burrows does not come without risk; the western spadefoot could be using the disturbed soil next to or immediately inside burrows to aid their digging, as has been found for the Great Basin spadefoot [14], without directly facing the occupants of the burrows.

The spadefoots in our study stayed closer to the pool locations compared to sympatric species such as the Baja California treefrog (*Pseudacris hypochondriaca*), western toad (*Anaxyrus boreas*), and *A. californiense*. Brattstrom and Warren [51] found *P. hypochondriaca* 457–914 m from a lake in southern California, whereas *A. boreas* have been found to move between an average of 218 m and 1800 m from breeding ponds depending on sex and site, and have been observed up to 7.4 km from their breeding sites [52–54]. *A. californiense* migrates farther than all but one other salamander species, with a median distance of 556 m [55]. By contrast, the spadefoots in our study moved a mean maximum distance of 69 m (SD ± 61.48) and a maximum distance of 262 m from the pool. Considering that we found rain to be a significant predictor of spadefoot movement and distance moved, the ongoing drought during our study could have negatively impacted spadefoot movement distances. Rainfall was about 50% below the 60-year average during the 2012–2013 season [42]. In wetter years, the western spadefoot could potentially move much longer distances. For comparison, the closely related eastern spadefoot (*S. holbrookii*) can disperse an estimated maximum distance 449 m away from breeding pools, though this is under conditions with four times the amount of rain that fell in southern California during our study [13], and the Great Basin spadefoot (*S. intermontane*) has been shown to move up to 2,350 m away from breeding pools [15],.

The movements we observed were not sufficient to connect the two sites studied to other known spadefoot breeding locations, the closest being a road rut 816 m from the Crystal Cove site. Knowledge of spadefoot dispersal is important for preserving the genetic diversity of *S. hammondii* populations [2]. Although we do not know if *S. hammondii* populations historically functioned as a metapopulation (i.e. with some exchange of individuals between subpopulations leading to increased genetic diversity and the recolonization of breeding pools following local extinction events [56]), our findings on spadefoot dispersal suggest that subpopulations may no longer be connected. Five years of drought dried pools, and the subsequent lack of breeding, could heighten the likelihood of local extinction at both the Crystal Cove and the Laguna Coast Wilderness sites.

Unfortunately, our study had some limitations, including male bias, low sample size and the fact that our study sites were spatially close together on the coast. In addition, there is always the chance that implantation of radio transmitters could impact behavior. However, telemetry studies of other spadefoot species have not shown a significant effect of transmitters on spadefoot behavior [13–15]. Because the spadefoot in our study utilized their terrestrial habitat differently from closely related and sympatric species, further research is warranted to determine if this difference was a result of low rainfall, or if it only applies to coastal populations. Western spadefoots are found in a variety of habitats, including coastal sage scrub, chaparral, oak woodlands, grasslands, washes, and floodplains along the California coast through the Central Valley and into the Sierra Nevada foothills [18,23]. Comparing movement and burrow preference of inland and coastal populations as well as repeating the study with a larger, more balanced sample size over multiple years to capture the effect of changes in climate on habitat use and movement could provide additional insight on the natural history of this species while also informing land managers as to the terrestrial requirements of different spadefoot populations.

## Supporting information

**S1 Table. Spadefoot summary data.** Table giving the length, mass, sex, capture date, number of telemetry fixes, number of burrows used, maximum distance from pool in meters, mean distance between burrows in meters, standard deviation of distance between burrows, minimum convex polygon home range in square meters, percent grass in that home range, depth of burrow in centimeters, and site location of each animal in our study.
(XLS)

## Acknowledgments

We thank our field volunteers and the Serrano Animal Hospital, in particular, Dr S. Weldy, Dr K. Krause and Dr N. Beaudet. This manuscript was improved by comments from D. Sandquist, J. Solera, and J. Carroll. Any use of trade names or specific product is for descriptive purposes only and does not imply endorsement of the U. S. Government. This study is contribution 502 of the U. S. Geological Survey Amphibian Research and Monitoring Initiative (ARMI).

## Author Contributions

**Conceptualization:** Katherine L. Baumberger, M. V. Eitzel, Matthew E. Kirby, Michael H. Horn.

**Data curation:** Katherine L. Baumberger, M. V. Eitzel.

**Formal analysis:** Katherine L. Baumberger, M. V. Eitzel.

**Funding acquisition:** Katherine L. Baumberger.

**Investigation:** Katherine L. Baumberger, Matthew E. Kirby, Michael H. Horn.

**Methodology:** Katherine L. Baumberger, Matthew E. Kirby, Michael H. Horn.

**Project administration:** Katherine L. Baumberger.

**Resources:** Katherine L. Baumberger.

**Supervision:** Katherine L. Baumberger.

**Visualization:** Katherine L. Baumberger.

Writing – **original draft:** Katherine L. Baumberger, M. V. Eitzel.

Writing – **review & editing:** Katherine L. Baumberger, M. V. Eitzel, Matthew E. Kirby, Michael H. Horn.

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
