## [Decision Letter · Decision Letter 0]

11 Jul 2019

PONE-D-19-16369

Movement and habitat selection of the western spadefoot (Spea hammondii) in southern California

PLOS ONE

Dear Ms. Baumberger,

Thank you for submitting your manuscript to PLOS ONE. After careful consideration, we feel that it has merit but does not fully meet PLOS ONE’s publication criteria as it currently stands. Therefore, we invite you to submit a revised version of the manuscript that addresses the points raised during the review process.

This is an interesting study with important conservation implications. Both reviewers and I agree that it is generally well written, and with some minor revisions, could be acceptable for publication. Please address all comments from both reviewers, as well as my few comments (see below) when revising your manuscript.

We would appreciate receiving your revised manuscript by Aug 25 2019 11:59PM. To enhance the reproducibility of your results, we recommend that if applicable you deposit your laboratory protocols in protocols.io, where a protocol can be assigned its own identifier (DOI) such that it can be cited independently in the future. For instructions see: http://journals.plos.org/plosone/s/submission-guidelines#loc-laboratory-protocols

We look forward to receiving your revised manuscript.

Kind regards,

William David Halliday, Ph.D.

Academic Editor

PLOS ONE

Journal Requirements:

2. We note that Figure 1 in your submission contain [map/satellite] images which may be copyrighted. All PLOS content is published under the Creative Commons Attribution License (CC BY 4.0), which means that the manuscript, images, and Supporting Information files will be freely available online, and any third party is permitted to access, download, copy, distribute, and use these materials in any way, even commercially, with proper attribution. For these reasons, we cannot publish previously copyrighted maps or satellite images created using proprietary data, such as Google software (Google Maps, Street View, and Earth). For more information, see our copyright guidelines: http://journals.plos.org/plosone/s/licenses-and-copyright.

You may seek permission from the original copyright holder of Figure 1 to publish the content specifically under the CC BY 4.0 license. 

If you are unable to obtain permission from the original copyright holder to publish these figures under the CC BY 4.0 license or if the copyright holder’s requirements are incompatible with the CC BY 4.0 license, please either i) remove the figure or ii) supply a replacement figure that complies with the CC BY 4.0 license. Please check copyright information on all replacement figures and update the figure caption with source information. If applicable, please specify in the figure caption text when a figure is similar but not identical to the original image and is therefore for illustrative purposes only.

Additional Editor Comments:

This is an interesting study with important conservation implications. Both reviewers and I agree that it is generally well written, and with some minor revisions, could be acceptable for publication. Please address all comments from both reviewers, as well as my few comments (see below) when revising your manuscript.

Minor comments:

Manuscript currently uses both “vernal” and “ephemeral”. Both are synonymous, but better to use consistent terminology throughout, so pick one and stick with it.

Line 73: Should say “and the species is under review” (not “in under review”)

The use of percent in the results makes the text quite confusing. I’ll use results from the Burrow Locations as an example. How can 100% duff be selected 95% of the time and 100% grass selected 32% of the time? The spadefoot would spend > 100% of time in both habitats, which is not possible, unless of course a site can be both 100% grass and 100% duff. Similarly, spadefoots select sites without pre-existing burrows 43% of the time, but sites with pre-existing burrows 76% of the time. In this latter example, I don’t know how the sum of sites without burrows and sites with burrows can add up to anything more than 100%. I assume you’re calculating log-odds ratios or something like this with your binomial regression results, but this is not equivalent to a percent. Please either clarify the interpretation of these results, or find a different way of communicating your results using something other than percent.

Line 352: add a comma or semi-colon before “very little native grass”. Otherwise, the sentence does not work.

Reviewers' comments:

Reviewer's Responses to Questions

**Comments to the Author**

1. Is the manuscript technically sound, and do the data support the conclusions?

Reviewer #1: Yes

Reviewer #2: Partly

2. Has the statistical analysis been performed appropriately and rigorously? 

Reviewer #1: Yes

Reviewer #2: Yes

3. Have the authors made all data underlying the findings in their manuscript fully available?

Reviewer #1: Yes

Reviewer #2: Yes

4. Is the manuscript presented in an intelligible fashion and written in standard English?

Reviewer #1: Yes

Reviewer #2: Yes

5. Review Comments to the Author

Reviewer #1: This study examines the upland habitat use of western spadefoot toads. Overall, statistical analyses seem sound, and it provides a solid baseline of habitat use data. As you will see, my only major concern is the description of the principal components analysis and its usefulness for guiding conservation. I provide a suggestion for replacing it below.

Lines 44-46: It would be shorter to write “range: 1-204 m” than to write out “min = 1 m – max = 204 m”

Line 73: Change “in” to “is”

Line 74: Capitalize Endangered or Threatened

Line 126: Can you say a bit more about what the animals were doing when captured? Were they migrating toward or away from the breeding pond, etc.?

Line 135: Place apostrophe after the s and change location to locations.

Line 146: Slope would not be recorded with a compass. How was slope measured?

Lines 208-210, 310-313 & 321-323: The authors need to change the way they describe principal components. The first principal component is not the average of the sites. The first principal component is the primary axis of variation across the sites. The second principal component captures as much of the remaining variation as possible, provided that it is orthogonal to the first. If the authors decide to keep the principal components analysis, then they should report the percentage of variation that is captured by each principal component. However, I would suggest dropping the principal components entirely. I realize that the reason for using them was that the three axes in the ternary graph are not independent from each other, and thus there was an issue with analyzing all three. However, the principal components in this case are difficult to interpret (they almost always are), so the reader is left wondering, “What does it mean that toads have a negative association with principal component 2?” It sounds like the interpretation is that toads dislike clay and like sand? Why not just analyze those two axes of the ternary plot and skip silt? That way you are not analyzing the third confounded axis, and if there is a direct association between toads and either clay or sand, then readers will be able to interpret what that actually means and apply it to conserving appropriate habitat for spadefoots.

Line 232: Insert comma after “period”

Line 235: Insert comma before “with”

Line 237: Insert comma before “which”

Line 250: In Table 2 it gives a wide range of MCPs, making it seem like MCPs were created for each individual toad. In the text, however, it only describes methods in which a single MCP was created for all toads at the same site. Can the methods be updated?

Lines 261-263: The movement distances predicted by the model seem high given that the mean distance between burrows was 18 m. Why are all predicted movements above the mean movement distance?

Lines 282-284: Why is this result not in Table 3? This seems like one of the most important results, and currently if a reader only looked at Table 3 they would think that toads are avoiding grasses rather than usually inhabiting them. I realize those results are at two different spatial scales, but someone skimming the paper might miss that distinction unless they are both highlighted in the table.

Lines 349-354: This section is trying to explain why western spadefoots don’t prefer bare ground the way that two of their close relatives do. However, bare ground wasn’t even one of the vegetation cover categories in this dataset. Thus, western spadefoots might prefer bare ground if it was available, but it simply isn’t available at these sites? Among the vegetation cover types that are available, is duff most similar to bare ground? Also, insert a semi-colon before “very”.

Lines 357-358: If tree cover is truly important, then why isn’t there a positive association with leaf litter?

Line 375: Insert a comma before with.

Lines 384-385: I think it’s good to bring up the possibility of limited rainfall being a confounding factor. However, I don’t know that is should be dwelt on quite this much. Eastern spadefoot are a different species, and thus may migrate a different distance for a number of reasons. Maybe drop this sentence?

Line 393: Insert comma after pools.

Line 394: Change heighten to heightened.

Reviewer #2: Manuscript Title: Movement and habitat selection of the western spadefoot (Spea hammondii) in southern California.

Manuscript Number: PONE-D-19-16369

Reviewer Comments to Authors:

This work describes research competed for the western spadefoot (Spea hammondii) on burrow selection and upland habitat use for two discrete sites within southern California. Utilizing radio-telemetry, the authors tracked 15 animals to determine the selection of burrow sites and the habitat preferences at these sites. The authors note that movements away from breeding ponds and between burrow sites are relatively small, restricted to less than 262-m. Western spadefoot in this study appear to select for specific habitat attributes at the microhabitat scale including selection for duff over grass or shrub cover. This study provides the first work on burrow habitat selection for western spadefoot.

Overall the manuscript is relatively clear and concise and provides an excellent starting point for further research for this species. While no major issues were noted during this review, several minor issues were noted that are highlighted below and annotated in the .pdf of the manuscript which is included in this review submission. Note that the annotated .pdf provides further comments and notes.

1. There are several grammatical issues that can be corrected to improve on the manuscript. Some of these have been noted in the marked up .pdf.

2. The introduction could benefit from a more comprehensive review of available literature for amphibian movements, in particular, an overview of available literature for other spadefoot and/or sympatric species.

3. A stronger argument could be developed for why this research is required. How does improving on the understanding of habitat use for western spadefoot support the management for this species? Your executive summary is good and provides some context that could be used here. There is certainly some good information here that could be expanded upon.

4. The description of the study area is good. The addition of some spatial reference would help to improve on this description.

5. The methods regarding radio telemetry would benefit from some additional clarifications including information regarding sexing of spadefoots, transmitter weight ratios and transmitter recovery.

6. The methods describing vegetation characteristics provide a good overview with some clarifications provided. In particular, Table 1 appears to provide results versus methods. Consideration for edits to this table could be made.

7. The description of methods for describing soil characteristics is also good with minor clarifications on core sample locations required.

8. The data analysis utilized for this work appears to be appropriate for the intent of the work.

9. The discussion section would benefit from a more through review of available literature to provide context for the results of this work.

10. The paper would benefit from a discussion on the limitations of the work. Specifically on the small, male biased, sample size and the limitations imposed by two sample sites. It would be useful to address any potential limitations or impacts to behaviour resulting from surgical implantation of transmitters.

11. The paper would benefit from a more thorough discussion on potential future research including expanding on sample sizes and the number of study sites.

6. PLOS authors have the option to publish the peer review history of their article (what does this mean?). If published, this will include your full peer review and any attached files.

Reviewer #1: No

Reviewer #2: Yes: Dustin Oaten

---

## [Author Response · Author response to Decision Letter 0]

24 Aug 2019

http://www.journals.plos.org/ and http://www.journals.plos.org/

We have made our best effort to comply with the style requirements, including file naming.

2. We note that Figure 1 in your submission contain [map/satellite] images which may be copyrighted. All PLOS content is published under the Creative Commons Attribution License (CC BY 4.0), which means that the manuscript, images, and Supporting Information files will be freely available online, and any third party is permitted to access, download, copy, distribute, and use these materials in any way, even commercially, with proper attribution. For these reasons, we cannot publish previously copyrighted maps or satellite images created using proprietary data, such as Google software (Google Maps, Street View, and Earth). For more information, see our copyright guidelines: http://journals.plos.org/.

We require you to either (1) present written permission from the copyright holder to publish these figures specifically under the CC BY 4.0 license, or (2) remove the figures from your submission

We have replaced the copyrighted base map with a NRCS/USDA base map that has no use constraints. We have credited the originator in the figure. The disclaimer can be found here: https://datagateway.nrcs.usda.gov/disclaimer.html. 

Additional Editor Comments:

This is an interesting study with important conservation implications. Both reviewers and I agree that it is generally well written, and with some minor revisions, could be acceptable for publication. Please address all comments from both reviewers, as well as my few comments (see below) when revising your manuscript.

Minor comments:

Manuscript currently uses both “vernal” and “ephemeral”. Both are synonymous, but better to use consistent terminology throughout, so pick one and stick with it.

Change accepted

Line 73: Should say “and the species is under review” (not “in under review”)

Change accepted

The use of percent in the results makes the text quite confusing. I’ll use results from the Burrow Locations as an example. How can 100% duff be selected 95% of the time and 100% grass selected 32% of the time? The spadefoot would spend > 100% of time in both habitats, which is not possible, unless of course a site can be both 100% grass and 100% duff. Similarly, spadefoots select sites without pre-existing burrows 43% of the time, but sites with pre-existing burrows 76% of the time. In this latter eWe appreciate the attention to the flow of the introduction xample, I don’t know how the sum of sites without burrows and sites with burrows can add up to anything more than 100%. I assume you’re calculating log-odds ratios or something like this with your binomial regression results, but this is not equivalent to a percent. Please either clarify the interpretation of these results, or find a different way of communicating your results using something other than percent.

Thank you for noting the confusing nature of these sections. It is of course challenging to present results on the probability scale which are meaningful to the question at hand (especially when some of the predictor variables are themselves percents!). We have re-written the section to clarify that it is not whether a given novel spadefoot chooses A or B (because they must have chosen one or the other, i.e. the probabilities sum to 1), but rather the probability that a spadefoot encountering a specific novel site with given characteristics will choose to burrow there or not. We have also shifted to referring to probabilities rather than percentages.

For example, “the model predicts that a spadefoot encountering a site composed entirely of grass has probability 0.32 of choosing to burrow there (significantly less than 0.50); if the site were composed entirely of shrub, a probability 0.42 of choosing to burrow there (significantly less than 0.50); and if it were composed entirely of duff, a probability 0.95 of choosing to burrow (significantly more than 0.50).” We have similarly rephrased the other parts of the results referring to percentages/probabilities as well. 

Line 352: add a comma or semi-colon before “very little native grass”. Otherwise, the sentence does not work.

Change accepted

Reviewer Comments to Authors:

Reviewer #1: This study examines the upland habitat use of western spadefoot toads. Overall, statistical analyses seem sound, and it provides a solid baseline of habitat use data. As you will see, my only major concern is the description of the principal components analysis and its usefulness for guiding conservation. I provide a suggestion for replacing it below.

Lines 44-46: It would be shorter to write “range: 1-204 m” than to write out “min = 1 m – max = 204 m”

Change accepted

Line 73: Change “in” to “is”

Change accepted

Line 74: Capitalize Endangered or Threatened

Change accepted

Line 126: Can you say a bit more about what the animals were doing when captured? Were they migrating toward or away from the breeding pond, etc.?

We have added this information. Seven of the animals were found on the surface, presumably foraging since that breeding pond did not fill. The other eight were found in or near the breeding pond when water was present.

Line 135: Place apostrophe after the s and change location to locations.

Change accepted

Line 146: Slope would not be recorded with a compass. How was slope measured?

Slope was measured with the clinometer that is built into the compass. This has been added to the manuscript.

Lines 208-210, 310-313 & 321-323: The authors need to change the way they describe principal components. The first principal component is not the average of the sites. The first principal component is the primary axis of variation across the sites. The second principal component captures as much of the remaining variation as possible, provided that it is orthogonal to the first. If the authors decide to keep the principal components analysis, then they should report the percentage of variation that is captured by each principal component. However, I would suggest dropping the principal components entirely. I realize that the reason for using them was that the three axes in the ternary graph are not independent from each other, and thus there was an issue with analyzing all three. However, the principal components in this case are difficult to interpret (they almost always are), so the reader is left wondering, “What does it mean that toads have a negative association with principal component 2?” It sounds like the interpretation is that toads dislike clay and like sand? Why not just analyze those two axes of the ternary plot and skip silt? That way you are not analyzing the third confounded axis, and if there is a direct association between toads and either clay or sand, then readers will be able to interpret what that actually means and apply it to conserving appropriate habitat for spadefoots.

We have corrected the description of the principal components and included the percent of variation accounted for in each component. The methods section now states: “Therefore we conducted a principal components analysis (using the “prcomp” function in R), which gave a first component reflecting the greatest variation in soil composition in our sites (98%), and a second component orthogonal to the first which reflected the second greatest variation in soil characteristics (2%).” We have similarly corrected the language elsewhere in the manuscript to match, and have also made a note in the Results that the spadefoots do not apparently respond to the axis representing the most variation, but to the second component orthogonal to that.

The principal components are actually in some ways more interpretable than the original variables, because the three variables are highly correlated (in addition to being complementary/summing to 100%). We have added a note to the Methods regarding the correlations, as well as the fact that our sites appear to be confined to a narrow band of soil textures. 

Note that trying to use the raw variables has poor results due to the correlations between them. However, we do agree with Reviewer #1 that principal components are difficult to draw conclusions from. Because only one component was significant in the statistical model, we give the coefficients from that component (PC2) and extrapolate from the coefficients and the sign of its model parameter that spadefoot prefer sand and silt and dislike clay (which is consistent with what one can see in the figure). We also report parameters in the results for a parallel model using silt and clay (the two least correlated soil textures), which confirms that the preference against clay is stronger than the preference for silt.

Line 232: Insert comma after “period”

Change accepted

Line 235: Insert comma before “with”

Change accepted

Line 237: Insert comma before “which”

Change accepted

Line 250: In Table 2 it gives a wide range of MCPs, making it seem like MCPs were created for each individual toad. In the text, however, it only describes methods in which a single MCP was created for all toads at the same site. Can the methods be updated?

The methods have been updated. We had originally only included minimum convex polygons by site because we had so few locations, but we estimated them by animal to show the variation in the amount moved.

Lines 261-263: The movement distances predicted by the model seem high given that the mean distance between burrows was 18 m. Why are all predicted movements above the mean movement distance?

The movement distances predicted by the model are for radial distances from the pool (mean distance = 40 m), while the mean distance between burrows could be much shorter. We have added some text to clarify this: “Number of meters moved away from or towards the breeding pool was predicted by rain in cm and by the animal random effect.”

Lines 282-284: Why is this result not in Table 3? This seems like one of the most important results, and currently if a reader only looked at Table 3 they would think that toads are avoiding grasses rather than usually inhabiting them. I realize those results are at two different spatial scales, but someone skimming the paper might miss that distinction unless they are both highlighted in the table.

Table 3 includes only statistical results, and the results pertaining to the remotely sensed data were not statistically tested in the previous draft, only summarized. We have now included a Chi-squared test of independence for these results in Table 3 (see row 1 and footnote 2). Note that we have now simplified the text in the prose portion of the results and referred to the Table. (We have also mentioned the Chi-squared test in the Methods)

Lines 349-354: This section is trying to explain why western spadefoots don’t prefer bare ground the way that two of their close relatives do. However, bare ground wasn’t even one of the vegetation cover categories in this dataset. Thus, western spadefoots might prefer bare ground if it was available, but it simply isn’t available at these sites? Among the vegetation cover types that are available, is duff most similar to bare ground? 

We had initially not included bare ground in the analysis, because it was the complement of grass, leaf litter, and duff (they typically summed to 100%, constituting the understory layer). This means that bare ground percentage is strongly correlated with those variables and will impact their parameter estimates and p-values, so we chose to leave it out of the statistical model. We had also left out tree cover, because tree cover was rare in our dataset.

We have re-done the statistical model to include tree cover and bare/open ground, and neither are significant (the significance and relative magnitude of the parameters for the other cover types are essentially unchanged). We now report tree and bare ground cover statistical results in Table 3, along with slightly updated parameter estimates and p-values for grass, duff, and shrub. 

Also, insert a semi-colon before “very”.

Change accepted.

Lines 357-358: If tree cover is truly important, then why isn’t there a positive association with leaf litter?

Leaf litter is transported some distance away from a given tree and could serve a different function for spadefoots. We highlight the other ecological functions of trees as a potentially important habitat factor: shade, hydraulic lift. 

Line 375: Insert a comma before with.

Change accepted.

Lines 384-385: I think it’s good to bring up the possibility of limited rainfall being a confounding factor. However, I don’t know that is should be dwelt on quite this much. Eastern spadefoot are a different species, and thus may migrate a different distance for a number of reasons. Maybe drop this sentence?

We do think that rain has a significant impact on spadefoot movement, but have taken this comment into consideration and dropped the sentence.

Line 393: Insert comma after pools.

Change accepted

Line 394: Change heighten to heightened.

We are not certain that it has heightened the risk, therefore we changed this to “could heighten.”

Reviewer #2: 

This work describes research competed for the western spadefoot (Spea hammondii) on burrow selection and upland habitat use for two discrete sites within southern California. Utilizing radio-telemetry, the authors tracked 15 animals to determine the selection of burrow sites and the habitat preferences at these sites. The authors note that movements away from breeding ponds and between burrow sites are relatively small, restricted to less than 262-m. Western spadefoot in this study appear to select for specific habitat attributes at the microhabitat scale including selection for duff over grass or shrub cover. This study provides the first work on burrow habitat selection for western spadefoot.

Overall the manuscript is relatively clear and concise and provides an excellent starting point for further research for this species. While no major issues were noted during this review, several minor issues were noted that are highlighted below and annotated in the .pdf of the manuscript which is included in this review submission. Note that the annotated .pdf provides further comments and notes.

1. There are several grammatical issues that can be corrected to improve on the manuscript. Some of these have been noted in the marked up .pdf.

See responses below.

2. The introduction could benefit from a more comprehensive review of available literature for amphibian movements, in particular, an overview of available literature for other spadefoot and/or sympatric species.

We have cited several papers on other spadefoot species elsewhere in the paper, but we have now rewritten the first two paragraphs of the introduction and now include those citations here as well. We have also included new citations (Richardson and Oaten 2013).

Richardson, J.S. et D. Oaten. 2013. Critical breeding, foraging, and overwintering habitats of Great Basin spadefoot toads (Spea intermontana) and western toads (Anaxyrus boreas) within grassland ecosystems: 2013 final report. Prepared for Can. Wildl. Fed., Kanata, ON.

3. A stronger argument could be developed for why this research is required. How does improving on the understanding of habitat use for western spadefoot support the management for this species? Your executive summary is good and provides some context that could be used here. There is certainly some good information here that could be expanded upon.

We have rewritten the first two paragraphs of the introduction to clarify the argument for this study. We focus on the natural history of burrowing species, where the distance from the pools is particularly important, and this information (as well as habitat preferences, etc) is not known for Western spadefoot.

4. The description of the study area is good. The addition of some spatial reference would help to improve on this description.

See below (spatial reference to entrance gates added).

5. The methods regarding radio telemetry would benefit from some additional clarifications including information regarding sexing of spadefoots, transmitter weight ratios and transmitter recovery.

See below (this information was added)

6. The methods describing vegetation characteristics provide a good overview with some clarifications provided. In particular, Table 1 appears to provide results versus methods. Consideration for edits to this table could be made.

Table 1 has been moved to the results section and has been renumbered as Table 2.

7. The description of methods for describing soil characteristics is also good with minor clarifications on core sample locations required.

We have added language to clarify the locations of core samples.

8. The data analysis utilized for this work appears to be appropriate for the intent of the work.

9. The discussion section would benefit from a more through review of available literature to provide context for the results of this work.

We have conducted another literature review and added several citations (see below).

10. The paper would benefit from a discussion on the limitations of the work. Specifically on the small, male biased, sample size and the limitations imposed by two sample sites. It would be useful to address any potential limitations or impacts to behaviour resulting from surgical implantation of transmitters.

We have added language discussing the limitations of the work. Unfortunately, there is no way to do this kind of study without telemetry and there is always the possibility that any kind of transmitter will have an impact on animal behavior. But previous studies have found no significant effect and we have inserted language to reflect that.

11. The paper would benefit from a more thorough discussion on potential future research including expanding on sample sizes and the number of study sites.

We have added language addressing the need for further studies with larger sample sizes and more study sites in different habitats.

Specific comments from pdf:

line 57: “There is information available for the Great Basin spadefoot showing movements away from breeding ponds ”

The introduction now includes references to maximum distances for other spadefoot species (including Great Basin spadefoot).

line 64: “A stronger argument could be developed for why this research is required. How does improving on habitat use understanding support the management for this species and this habitats? Your executive summary is good and provides some context that could be used here.”

See response to general comments above, point 3

line 67: “Is there a more fluid way to tie these together? ”

We appreciate the attention to the flow of the introduction – we have now edited the introduction to clarify the importance of the study, which means the logic proceeds from buffer widths to the distances of aestivation of fossorial species, and from there to habitat use and movement. We have omitted the details regarding California regulations and species of special concern (this material is treated in the discussion).

line 74: “Italicize.” (ESA)

PLoS style is not to italicize "Endangered Species Act," see for example:

Dunk JR, Woodbridge B, Schumaker N, Glenn EM, White B, LaPlante DW, et al. (2019) Conservation planning for species recovery under the Endangered Species Act: A case study with the Northern Spotted Owl. PLoS ONE 14(1): e0210643. https://doi.org/10.1371/journal.pone.0210643

Valdivia A, Wolf S, Suckling K (2019) Marine mammals and sea turtles listed under the U.S. Endangered Species Act are recovering. PLoS ONE 14(1): e0210164. https://doi.org/10.1371/journal.pone.0210164

Gibbs KE, Currie DJ (2012) Protecting Endangered Species: Do the Main Legislative Tools Work? PLoS ONE 7(5): e35730. https://doi.org/10.1371/journal.pone.0035730

line 80: “size”

Change accepted

line 104: “UTMs or other spatial reference.”

Because of the sensitivity of the species we do not provide the location of the breeding pools; instead we provide coordinates for the entrance gates of the parks which are publicly accessible.

line 126: “Reference”

A reference was added.

line 126: “Include info on morphological features used to determine sex.”

Change accepted, we have added language to describe how we determined the animal’s sex.

line 127: “Were reproductive females excluded from captured animals?”

Reproductive females were not excluded from captured animals, we just didn’t catch very many of them. As stated, we captured animals opportunistically, and calling males are easier to find. 

line 129: “Weight of transmitter and relative proportion of study animal mass.”

We have added language with this information to the manuscript.

line 132: “How are ill effects defined? Proportion of body mass loss, maintenance of capture weight? ”

We have added language with this information to the manuscript.

line 137: “Should explicitly state here that transmitters were not removed. Also note why utilizing implanted transmitters was used.”

We have added language about why we used internal transmitter. We state that transmitters were not removed in the results section (line 257)

line 142: “State explicitly how many plots were completed (167?) .”

We have explicitly stated how many plots were completed. We feel this fits better into the results section; it has been added there.

line 150: “Noted as 262-m elsewhere.”

We used a 300 m radius to encompass the maximum single movement of the spadefoot, plus a bit extra in cased they moved some more. We added language to this effect.

line 162: “Title for the table could be updated to reflect the information presented. This is a summary of the characteristics at burrow locations. The rainfall data is assumed to be for each site, versus burrow locations. ”

Change accepted.

line 162: “This Table seems to be better suited to being placed in the results section versus material and methods.” (Table 1)

Change accepted.

line 168: “How many different spadefoots did this represent? Need to understand if this data was collected from several representative burrow locations,”

We have included language to make it clear that all 15 spadefoot were included with 11 samples.

line 172: “Table 1, above, does not appear to depict burrow depth.”

We have renumbered the tables. Table 1 now depicts burrow depth.

line 247: “Remove” (species name)

Change accepted

line 327: “Specifically burrow site characteristics.”

Change accepted.

line 368: “Garner (2012) notes that Great Basin spadefoot utilized the friable soil inside animal burrows as a starting point for burrow sites.”

We have added this reference.

line 373: “There are several research articles on western toad that highlight much longer movements away from breeding ponds (>7 km).”

We have added a reference to this effect. 

Schmetterling DA, Young MK. Summer movements of boreal toads (Bufo boreas boreas) in two western Montana basins. J Herptol. 2008;42:111-123.

line 376: “Note that there is recent information for the Great Basin spadefoot showing much larger movements away from breeding ponds - up to 2,350 m (see "Recovery Strategy for the Great Basin Spadefoot (Spea intermontana) in Canada' for a summary).” 

Environment and Climate Change Canada. 2017. Recovery Strategy for the Great Basin Spadefoot (Spea intermontana) in Canada. Species at Risk Act Recovery strategy series. Environment and Climate Change Canada, Ottawa. 2 parts, 31 pp. + 40 pp.

We have added this to the manuscript for comparison.

line 380: “Good point. Do note that spadefoot toads are very well adapted to dry ecosystems and rainfall may not be restrictive to movements. ”

From the results of our study we feel that rainfall could be restrictive to movements. They have adapted to dry ecosystems by not moving when it doesn’t rain.

This figure would benefit from a legend to define the lines.

We respectfully decline this change. We feel that a legend would be too crowded with all the spadefoot lines, and that the caption adequately defines the lines.

---

## [Editor Report · Decision Letter 1]

3 Sep 2019

Movement and habitat selection of the western spadefoot (Spea hammondii) in southern California

PONE-D-19-16369R1

Dear Dr. Baumberger,

We are pleased to inform you that your manuscript has been judged scientifically suitable for publication and will be formally accepted for publication once it complies with all outstanding technical requirements.

With kind regards,

William David Halliday, Ph.D.

Academic Editor

PLOS ONE

Additional Editor Comments (optional):

Good job on the revisions.
---

## [Editor Report · Acceptance letter]

25 Sep 2019

PONE-D-19-16369R1 

Movement and habitat selection of the western spadefoot (Spea hammondii) in southern California 

Dear Dr. Baumberger:

I am pleased to inform you that your manuscript has been deemed suitable for publication in PLOS ONE. Congratulations! Your manuscript is now with our production department. 

With kind regards,

on behalf of

Dr. William David Halliday 

Academic Editor

PLOS ONE